# Computer-Aided Diagnosis Algorithm for Classification of Malignant Melanoma Using Deep Neural Networks

**DOI:** 10.3390/s21165551

**Published:** 2021-08-18

**Authors:** Chan-Il Kim, Seok-Min Hwang, Eun-Bin Park, Chang-Hee Won, Jong-Ha Lee

**Affiliations:** 1Department of Biomedical Engineering, Keimyung University, Daegu 42601, Korea; David.chanil.kim@gmail.com (C.-I.K.); meen2378@naver.com (S.-M.H.); peb0822@naver.com (E.-B.P.); 2Department of Electrical and Computer Engineering, Temple University, Philadelphia, PA 19122, USA; cwon@temple.edu

**Keywords:** malanoma, computer aided diagnosis, convolutional neural network

## Abstract

Malignant melanoma accounts for about 1–3% of all malignancies in the West, especially in the United States. More than 9000 people die each year. In general, it is difficult to characterize a skin lesion from a photograph. In this paper, we propose a deep learning-based computer-aided diagnostic algorithm for the classification of malignant melanoma and benign skin tumors from RGB channel skin images. The proposed deep learning model constitutes a tumor lesion segmentation model and a classification model of malignant melanoma. First, U-Net was used to classify skin lesions in dermoscopy images. We implement an algorithm to classify malignant melanoma and benign tumors using skin lesion images and expert labeling results from convolutional neural networks. The U-Net model achieved a dice similarity coefficient of 81.1% compared to the expert labeling results. The classification accuracy of malignant melanoma reached 80.06%. As a result, the proposed AI algorithm is expected to be utilized as a computer-aided diagnostic algorithm to help early detection of malignant melanoma.

## 1. Introduction

Malignant melanoma accounts for approximately 1–3% of all the cases of malignant tumors diagnosed in Western countries. The disease is particularly common among Caucasians, and its cases have been increasingly observed in South Korea as well [1]. In the United States, the International Skin Imaging Collaboration (ISIC) was established as an international cooperation organization to automatically analyze skin lesions, obtain relevant data, and expand research base for this field [2]. The ISIC has provided approximately 2000 images of malignant melanoma obtained from different clinical centers located in the world with professional diagnosis results. This study was conducted based on images and diagnosis results related to dermoscopy, which were provided by the ISIC in 2017 [3].

Malignant melanoma is a malignant skin tumor made of melanocytes. A melanocyte is a normal cell on the skin or mucous membrane that generates melanin on the skin. Malignant melanoma refers to cancer caused by such normal melanocytes. Malignant tumors generated on the skin include malignant melanoma, squamous cell carcinoma, and basal cell carcinoma. Among these diseases, malignant melanoma has the highest degree of malignancy. It is the most fatal form of skin cancer and accounts for approximately 75% of skin cancer deaths [4]. However, it is difficult to visually diagnose malignant melanoma because of its insignificant contrast effects against the skin and great similarity with benign skin tumors [5]. Therefore, a dermoscopy technique has been recently adopted in South Korea and other countries to increase the accuracy of skin cancer diagnosis.

Dermoscopy is a non-invasive skin imaging technique used to obtain enlarged skin images. Based on this technique, a skin lesion can be enlarged by approximately 10 times for observation. Dermoscopy is not used as a simple magnifier but as a tool that enables the user to observe the epidermis, the boundary between the epidermis and dermis, and the upper layer of the dermis through the stratum corneum by preventing light reflection on the skin surface based on mineral oil, alcohol, or water applied on a lesion [6]. Figure 1 shows dermoscopy images of malignant melanoma (Figure 1a) and seborrheic keratosis (Figure 1b), a type of a benign skin tumor.

An effective treatment method has not been developed to cure malignant melanoma, which has progressed to a certain degree. However, this disease can be completely treated when it is diagnosed in an initial stage; thus, it is crucial to accurately diagnose it [1]. For the diagnosis of malignant melanoma, cancer is generally observed with the naked eye or based on dermoscopy. When the lesion observed is suspected as malignant melanoma, a biopsy is carried out to derive more accurate diagnosis. Among the rules applied in the stage of observing a skin lesion for diagnosis, the ABCDE rule is as follows. First, A is an asymmetry, which represents asymmetrical properties. Second, B stands for border, which is irregular properties at the edge. C is color and it means diversity of color. D indicate diameter, the criterion for diagnosis based on diameter is 6 mm. The last E is evolution and represents change in size [7].

Malignant melanoma cannot be accurately diagnosed although it is observed based on dermoscopy and the ABCDE rule. Therefore, this study proposed an algorithm that can support an expert to analyze malignant melanoma based on a deep neural network (DNN). In this study, medical images of unspecified patients were utilized as dermoscopy images. Because these images were obtained from different distances, the D and E rules cannot be applied. To address this problem, this study conducted processes of splitting a lesion area and performing training of features from RGB lesion images according to the A, B, and C rules in a CNN [8].

Machine learning is a field of artificial intelligence where algorithms and techniques are developed to learn patterns and features from the data provided, independently define tasks, and perform tasks according to new data. It is also defined as a planning process for enabling a computer to exhibit optimal performance based on past experience such as sample data. For example, if a model consists of parameters, learning or training refers to an action of a computer program for optimizing the model parameters based on training data or past experience. Subsequently, the trained model can predict results from new data that were not provided in the training process. Machine learning has recently been applied to medical technologies including medical image analysis. It has also been widely used for overall medical image analyses, such as extraction and segmentation of organs or cancer parts from medical images, image matching, and image search.

In a previous study, a computer-aided decision support system for macro images taken with a general-purpose camera was proposed [9]. It is said in this paper that general imaging conditions are negatively affected by non-uniform illumination, which further affects the extraction of relevant information. To alleviate this, we use a multi-step illumination compensation approach to process the image to define a smooth illuminated surface and extract the infected area using the proposed multi-mode segmentation method. Lesion information was calculated as a set of features consisting of geometry, photometry, boundary series, and texture measurements. Information theory methods were used to reduce the redundancy of feature sets and model classification boundaries to differentiate between benign and malignant samples using support vector machine, random forest, neural network, and fast discriminant mixed member-based naive Bayesian classifiers.

Another study combined recent advances in deep learning with established machine learning approaches to create an ensemble of methods capable of segmenting skin lesions and proposed a system to analyze the detected area and surrounding tissues for melanoma detection [10]. The study evaluated using the largest public benchmark data set of dermoscopic images containing 900 training images and 379 test images and demonstrated a new state-of-the-art performance level with an area under the receiver operating characteristic curve of 7.5%, average precision was improved to 4%.

Existing computer-aided diagnosis methods exhibit limited performance for extracting optimal features and developing an appropriate algorithm for all the images because human involvement is required to perform medical image processing and implement a pattern recognition algorithm [11]. These methods can be properly applied when the image data are few or when only a certain image needs to be processed [12]. However, these methods cannot be utilized when medical images captured under the same conditions show different property cases [13].

The computer-aided diagnosis method based on deep learning, which was proposed in this study, can perform a training process by itself under the supervision of a human being using the concept of deeply staking artificial neural networks for fields, including feature extraction, lesion area extraction, and lesion classification [14]. To this end, U-Net [15] is utilized to segment a lesion area in a dermoscopy image and extract a lesion area from the original image. Subsequently, the CNN is utilized to predict malignant and benign tumors in the RGB image of the extracted lesion area.

## 2. Lesion Area Segmentation

### 2.1. Data

In this study, training sets of 2000 images among data provided by ISIC 2017 were utilized. A training set consists of a dermoscopy image (Figure 2a) showing a lesion, a binary splitting image (Figure 2b), which is a label for a correct answer, and a label for a correct answer provided by a medical expert for supervised learning in the CNN [16]. Binary splitting images provided by ISIC 2017 were manually segmented by medical experts. The datasets used in this study were classified in three cases: malignant melanoma, seborrheic keratosis, and moles [17]. Because this study focused on diagnosing malignant melanoma, cases of seborrheic keratosis and moles were analyzed as benign cases.

Among 2000 dermoscopy images used in this study, the images including an excessive amount of hair (Figure 3a), unnecessary marks for the training process of the model (Figure 3b,c), and a great amount of magnifier noise (Figure 3d) were removed from the datasets for deep learning. Consequently, 1161 images were utilized.

### 2.2. Preprocessing

The resolution of original images in the datasets varies; most images exhibit a high resolution of 1000 × 1000 or more. Therefore, it was deduced that the process where U-Net learned the original images was inefficient. Thus, original dermoscopy and binary splitting images were resized to have the resolution of 128 × 128. U-Net proposed by Ronneberger et al. [18] is a DNN model, which can perform semantic segmentation in an image. Because it conducts a training process based on unpadded convolution, the size of the output image according to the input image decreases in proportion to the number of convolutional layers. Thus, symmetric padding (Figure 4a) was applied to the dermoscopy images used in this study before they were trained by the model. Moreover, zero padding (Figure 4b) was applied to the binary splitting images before they were trained by the model. The dermoscopy images were also normalized to facilitate a smooth training process before being input in the model.

### 2.3. U-Net

U-Net, proposed by Ronneberger et al. [18] in 2015, is a U-shaped CNN used to perform semantic segmentation in medical images. According to Ronneberger et al. [18], the CNN is generally used to perform single class label classification for the output of an image. However, segmentation of a lesion area in a medical image includes regional information during image processing, requiring class labeling for each pixel [18]. Thus, the U-Net trained binary splitting images to distinguish lesion areas under the condition of supervised learning.

The softmax function used in the learning process is expressed based on Equation (1) and applied to derive an estimated value for the correct answer for the input image.
(1)$$p_{k}(x)=\exp(a_{k}(x))/\left(\sum_{k^{\prime }=1}^{K}\exp(a_{k^{\prime }}(x))\right)$$

The cross-entropy loss function for each correct answer pixel is expressed based on Equation (2). A weight, ω(x), is additionally applied to the existing cross entropy loss function.
(2)$$E=\sum_{x\in \Omega }\omega (x)\log(p_{\ell (x)}(x))$$

Here, ω(x) is a parameter that applies a weight to a pixel located at x among objects when several objects need to be segmented in an image. It can be calculated using Equation (3).
(3)$$\omega (x)=\omega_{c}(x)+\omega_{0}\exp\left(-\frac{{\left(d_{1}(x)+d_{2}(x)\right)}^{2}}{2\sigma^{2}}\right)$$

Here, $\omega_{c}(x)$ is determined according to the frequency of x; that is, it increases when adjacent pixels have the same class label. The exp equation includes a function ($d_{1}$), which indicates the distance of the lesion area located the closest to x and a function ($d_{2}$), denoting the distance of the lesion area located the second closest to x. This equation applies a higher weight as the gap between the pixels included in the lesion area decreases. In Equation (2), ℓ(x) of $p_{\ell (x)}$ is a function that returns a k value corresponding to the label of the correct answer in the softmax function. Therefore, the k value is returned to the log function in the cross-entropy loss function. Thus, the possibility of deriving a correct answer according to each pixel was derived to apply the weight of $\omega_{c}(x)$ and output the weight map presented in Figure 5. The weight map determined based on Equations (1)–(3) shows the number of weights to be applied for the learning process of U-Net.

It has also been enhanced based on overlapping and dense skip connections to address the need for more accurate segmentation in medical images. This allowed the model to more effectively capture the finer details of foreground objects when the high-resolution feature maps of the encoder network were progressively enhanced before being fused with the corresponding semantically rich feature maps of the decoder network. Fast delivery of high-resolution feature maps directly from the encoder to the decoder network, fusing semantically different feature maps [19].

It consists of an encoder and decoder connected via a series of nested dense convolution blocks. The main idea is to bridge the semantic gap between the functional maps of the encoder and decoder before fusion. For example, the semantic spacing between ($x^{0,0}$,$x^{1,3}$) is connected using a dense convolution block with three convolution layers.
(4)$$x^{i,j}=\{\begin{array}{cc} \gamma (x^{i-1,j}) & j=0\\ \gamma \left(\left[{\left[x^{i,k}\right]}_{k=0}^{j-1},\upsilon (x^{i+1,j-1})\right]\right), & j>0 \end{array}$$
where function γ is a convolution operation followed by an activation function and υ denotes an up-sampling layer. By default, a node at level *j* = 0 receives only one input from the previous layer. Two of the encoder nodes at level *j* = 1 receive inputs from the encoder subnetwork, but in two successive levels a node at level *j* > 1 receives *j* + 1 inputs, of which *j* inputs are from the previous *j* node. output. The last input in the same skip path is the up-sampled output from the lower skip path.

In addition, we suggest using deep supervision. Due to the nested skip paths, U-Net generates full resolution feature maps at multiple semantic levels that allow for in-depth supervision. A combination of binary cross entropy and dice coefficients was added as a loss function to each level of meaning.
(5)$$L(Y,\hat{Y})=-\frac{1}{N}\sum_{b-1}^{N}\left(\frac{1}{2}\cdot Y_{b}\cdot \log{\hat{Y}}_{b}+\frac{2\cdot Y_{b}\cdot {\hat{Y}}_{b}}{Y_{b}+{\hat{Y}}_{b}}\right)$$

${\hat{Y}}_{b}$ denote the flatten predicted probabilities and $Y_{b}$ denote the flatten ground truths of $b_{th}$ image respectively, and *N* indicates the batch size.

### 2.4. Learning by U-Net

The hyper-parameter conditions (Table 1) for U-Net used in this study and its structure and processing procedures (Figure 5) are as follows.

### 2.5. U-Net Training Results and Analysis

When symmetric padding is applied in the pre-processing stage, the skin lesion area in the image is partially padded at the edge of the image as if it were reflected on a mirror. Because the symmetric padding method can affect the image output by the U-Net model, the input images were set to 170 × 170 in size, considering the size of the output images for that of the input images. The dermoscopy images with the size of 170 × 170, which were pre-processed and output, were also processed by U-Net to be output in the size of 128 × 128. This size was consistent to that of the initially resized images. The symmetric padding area did not affect the output results. The segmentation performance of the proposed method was calculated based on a dice similarity coefficient. The calculation equation is expressed based on Equation (6). Here, *P* is a segmentation area predicted by the U-Net model, and G is a labeling segmentation area suggested by a medical expert. In the dice method, a value closer to 100 indicates more excellent segmentation performance.
(6)DICE(%)=(2×(P∩G)/(P+G))×100

Table 2 shows the ratio of the dice value based on all the datasets.

The average, maximum, and minimum dice values were 83.45%, 99.24%, and 9.58%, respectively. These values indicated that the segmentation performance in most images was satisfactory except in special cases. Figure 6 shows a dermoscopy image (Figure 6a), a binary splitting image with a dice value of 81.1% (Figure 6b), and an expert’s labeling image (Figure 6c).

In Figure 7, the lesion area (Figure 7a) extracted based on the expert labeling image is compared with the lesion area (Figure 7b) extracted based on the binary splitting image predicted by the U-Net model. Because the dice value was 81.1%, it was not analyzed as high. However, as indicated in Figure 7, the method based on the U-Net also extracted the lesion area precisely.

As shown in Figure 8, satisfactory segmentation performance was derived despite the presence of noise such as hair in the case (Figure 8a) where the lesion area was clearly contrasted from the skin in terms of color. However, the segmentation performance varied sensitively according to noise such as hair in the case (Figure 8b) where the lesion area was unlikely to be distinguished from the skin.

## 3. Classification of Lesion Segmented Images

### 3.1. Convolution Calculation

This study proposed a method that can extract an RGB lesion image from a dermoscopy image, learn features related to the A, B, and C rules, and assist a medical expert to diagnose malignant melanoma based on the data trained [20]. The CNN is a type of artificial neural network that performs convolution operations. Convolution calculation facilitates machine learning under the condition where spatial information on data is maintained. In the convolution process, a filter or kernel with a certain height and width is moved based on a specific stride and applied to the input data [21].

As introduced above, convolution is a process of extracting the features of an image. A feature map in the CNN refers to the result of adding bias to the output. The size of the matrix derived through convolution decreases as well as the resolution of the image derived through convolution. To create the resolution of the input and output images, zero padding is generally applied to the input image prior to the convolution process. Figure 9 shows a dermoscopy image (Figure 9a) and a feature map (Figure 9b,c). As shown in this figure, the feature map varies according to the kernel values applied to the image.

### 3.2. CNN

As described in Section 3.1, various feature maps can be output according to the kernel values. Because it is crucial to identify a feature map that is optimized to classify certain images, the CNN learns the parameter and bias values of the kernels that can result in displaying the optimized feature map for classification. A loss function defined in the CNN is consistent to the cross entropy loss function used in U-Net. However, the former adopts a method of comparing labels based on the image, unlike the latter, which adopts a method of comparing labels based on each pixel; that is, the former compares a dermoscopy image with a class such as a benign or malignant case.

### 3.3. Training by the CNN

In this study, the CNN model used four convolution layers and 64 kernels to output the feature map. Layers, such as max pooling and batch normalization [22], were added to facilitate a smooth training process of the model, and the ReLU function was applied as an activation function. Nine hundred and twenty-eight out of 1161 images were used as a training set for the model, and the other 233 images were used as a verification set in the training process.

The 1161 dermoscopy images and diagnosis results based on the images were used as input data in the CNN. Figure 10 briefly presents the CNN model used in this study.

### 3.4. CNN Training Results and Analysis

The training results showed that the accuracy based on the training set was 80.06% and that based on the verification set was 72.10%. To examine the training process of the model, the loss (Figure 11a) and accuracy graphs (Figure 11b) of the model were output and analyzed.

The biggest limitation that can be confirmed from the current experimental results seems to be the lack of the number of samples. In the existing CNN learning, techniques such as image augmentation exist to solve this problem, but they are not suitable for application to clinical data. Therefore, we concluded that it is necessary to obtain more data for CNN training in order to correct the experimental results.

As shown in Figure 11, the loss and accuracy graphs of the model seemed to converge and become accurate in the initial training stage. However, a change did not occur as the training was repeated for a certain period. Thus, the accuracy derived based on the training and verification sets in the proposed model cannot be regarded as the result of the training performed by the model.

## 4. Conclusions

This paper proposed a malignant melanoma classification algorithm based on deep learning. It is important to diagnose and treat malignant melanoma in the initial state; however, the identification of this disease only with the naked eye is limited. Therefore, it tends to be diagnosed after progressing to a certain degree. This study verified that a training method based on two deep learning models can contribute to the early diagnosis of malignant melanoma through the extraction and classification of lesion areas. Among the deep learning models, the result of performing a training process based on U-Net indicated that satisfactory segmentation performance was achieved in most images except for a few images where the lesion areas were unlikely to be defined through the application of dermoscopy.

Several researchers have developed models that exhibit satisfactory performance for analyzing malignant melanoma based on deep learning. Therefore, it is expected that more accurate training results can be derived through the application of models such as deep CNNs, which are characterized by the increased stacking of layers of the CNN and represented by ResNet and GoogleNet. In this paper, a limited number of samples were used and in the future work, we will increase the number of samples and compare the algorithm.

## Figures and Tables

**Figure 1 sensors-21-05551-f001:**
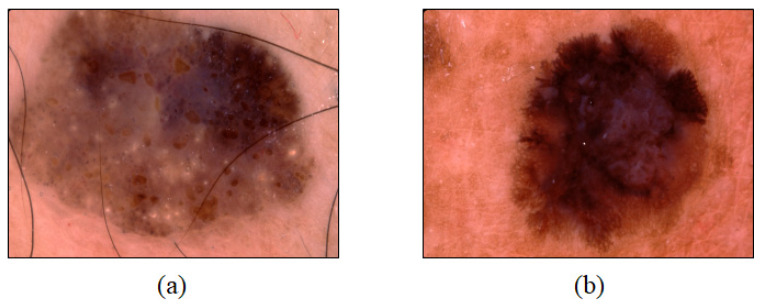
Dermoscopy images of (**a**) malignant melanoma and (**b**) a benign skin tumor (seborrheic keratosis).

**Figure 2 sensors-21-05551-f002:**
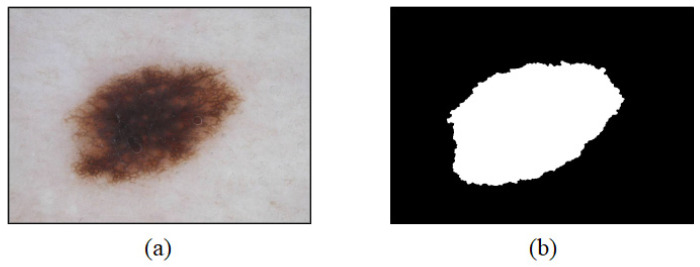
Examples of an image (**a**) obtained based on dermoscopy and an image and (**b**) obtained based on binary splitting.

**Figure 3 sensors-21-05551-f003:**
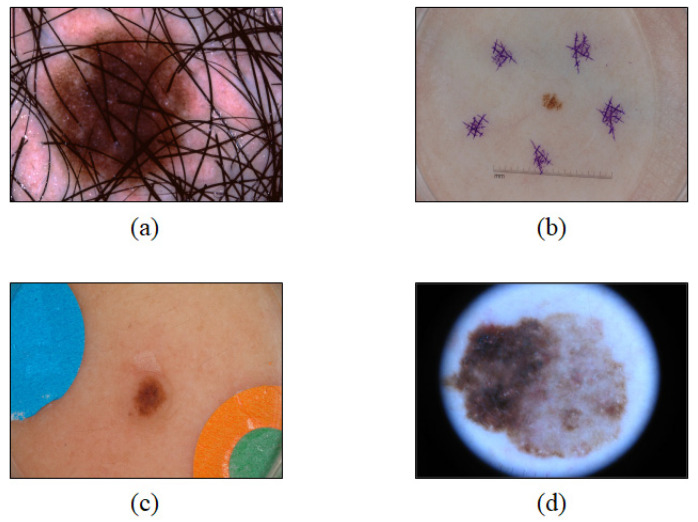
Examples of dermoscopy images excluded from the dataset: images including (**a**) an excessive amount of hair, (**b**) unnecessary marks, (**c**) an unnecessary mark, and (**d**) a great amount of noise.

**Figure 4 sensors-21-05551-f004:**
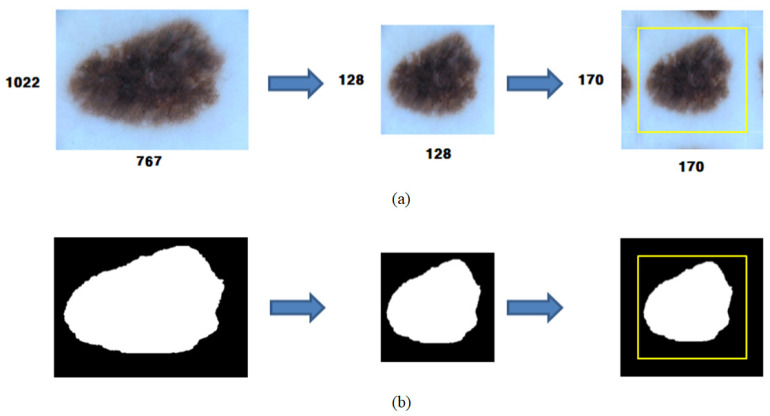
Examples of cases of applying (**a**) symmetric padding and (**b**) zero padding.

**Figure 5 sensors-21-05551-f005:**
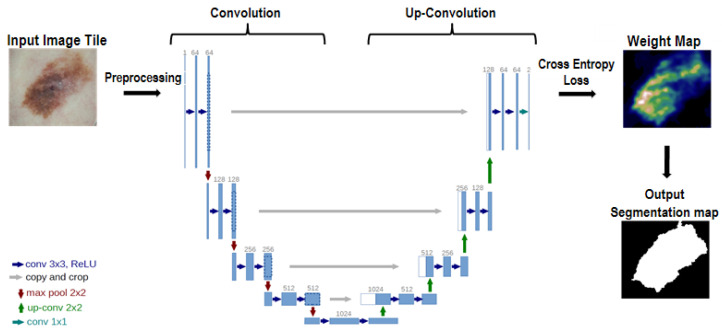
Structure and processing procedures of U-Net.

**Figure 6 sensors-21-05551-f006:**
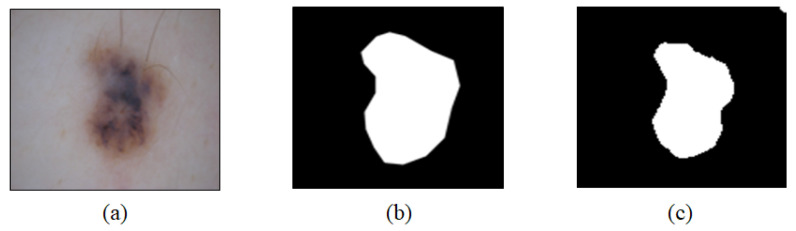
Results of professional labeling and U-Net images obtained based on a dermoscopy image. Images obtained based on (**a**) dermoscopy, (**b**) splitting image, and (**c**) labeling.

**Figure 7 sensors-21-05551-f007:**
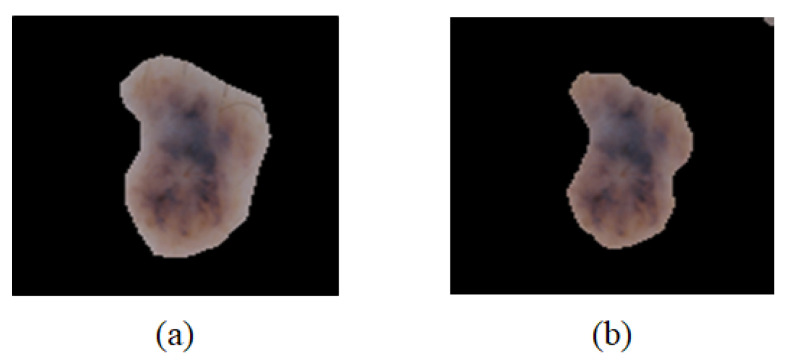
Images of skin lesion areas extracted based on the binary splitting image. (**a**) Labeling-based area and (**b**) predicted area based on U-Net.

**Figure 8 sensors-21-05551-f008:**
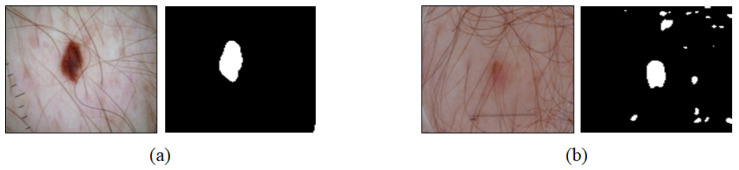
Examples of noise-robust and noise-vulnerable cases. (**a**) Lesion is clearly contrasted from the skin and (**b**) lesion is unlikely to be contrasted from the skin.

**Figure 9 sensors-21-05551-f009:**
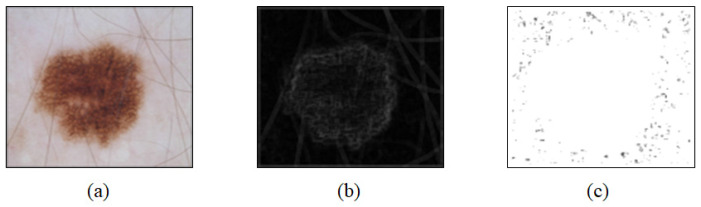
Examples of the image obtained based on dermoscopy and a feature map according to the dermoscopy image. (**a**) Dermoscopy image and (**b**,**c**) feature maps.

**Figure 10 sensors-21-05551-f010:**
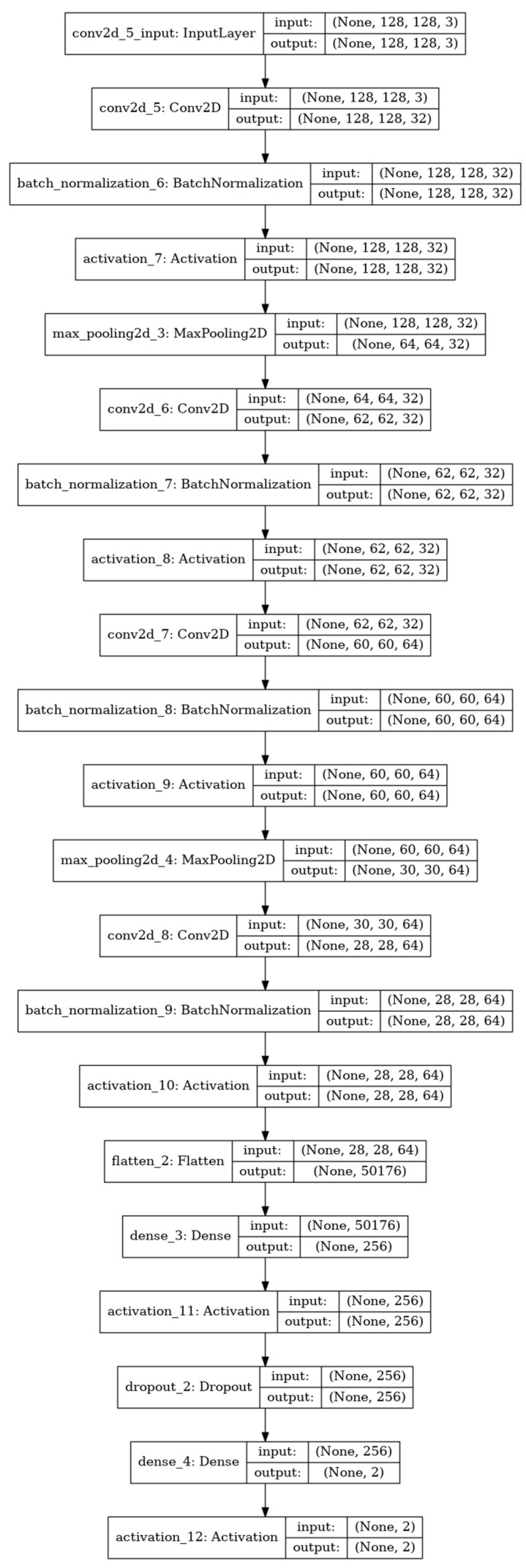
Summary of the CNN model.

**Figure 11 sensors-21-05551-f011:**
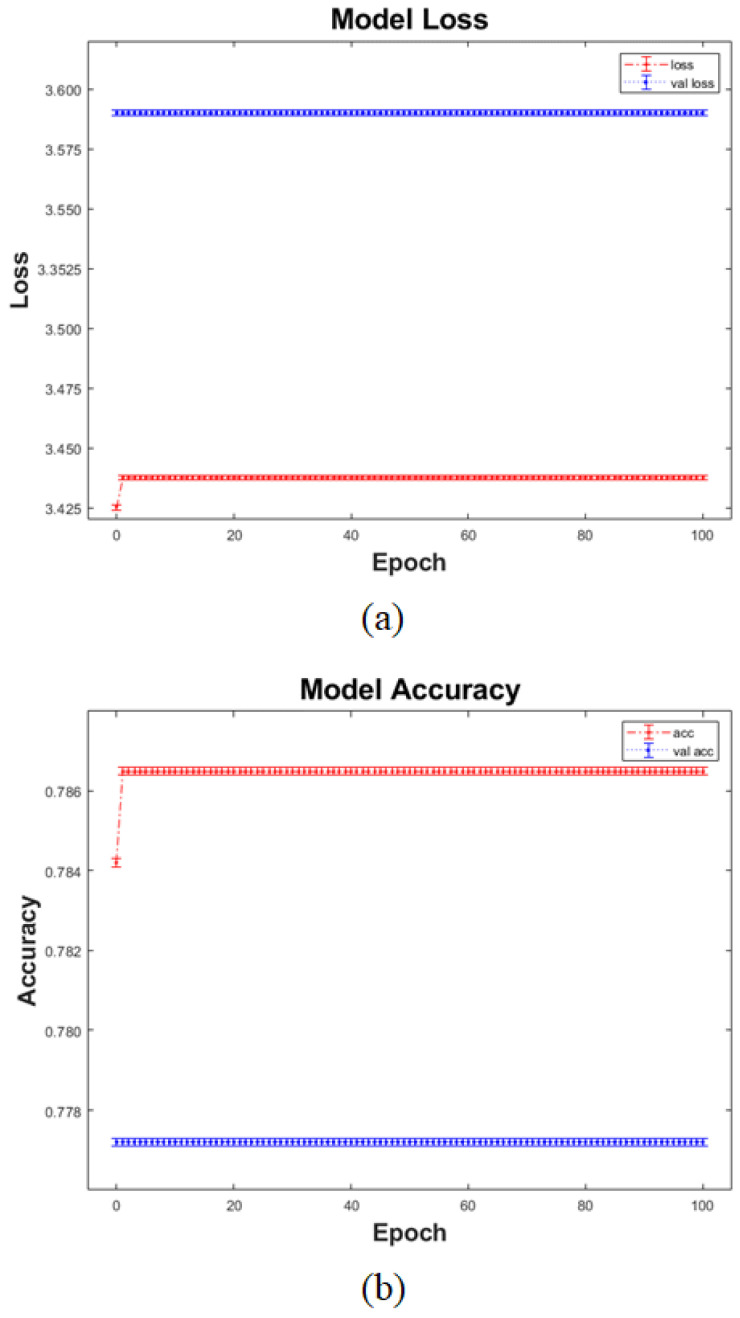
History graphs of the CNN model in terms of loss and accuracy. (**a**) Loss graph and (**b**) accuracy graph.

**Table 1 sensors-21-05551-t001:** Hyper-parameter conditions for U-Net.

Layer	3
Feature	64
Filter Size	3 × 3
Pool Size	2 × 2
Stride	1
Optimizer	‘Adam’

**Table 2 sensors-21-05551-t002:** Ratio of the dice value based on all the datasets.

Ratio of the Dice Value Based on All the Datasets (%)	
Ratio of the dice value at 90% or higher	42.2%
Ratio of the dice value at 80% or higher	25.8%
Ratio of the dice value at 70% or higher	17.4%
Ratio of the dice value at 70% or below	14.3%

## Data Availability

Not applicable.

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
