# Peer review of "Computer-Aided Diagnosis Algorithm for Classification of Malignant Melanoma Using Deep Neural Networks"

_sensors, 2021, doi:10.3390/s21165551_

Round 1
Reviewer 1 Report
The manuscript proposes a deep learning based method for malignant melanoma diagnosis. U-Net is applied to segment the lesion area, and CNN is applied for classification. Training and evaluation were performed on a data set with 1161 samples.
Major comments:
- The manuscript provides too many details about the development of Deep Learning. It’s better to cut down it to a brief introduction.
- In chapter 3.3 Training by the CNN, 928 samples were used to train the model. No augmentation strategies were mentioned. Considering the size of the CNN. The number of the samples may be insufficient.
- The analysis of the results shown in Figure 11 is sufficient. The straight lines occurred soon after training may be caused by insufficient training samples.
- There are no evaluation results on different data sets, and no comparison with other state-of-the-art methods.
Minor comments:
- Chapter 2.2 Preprocessing does not mention, when performing zero padding, what’s the size of the blank boundary.
- It’s not clear why the lesion area in Figure 6(a) and (c) looks different.
Author Response
Response to Reviewer 1 Comments
We are grateful for the reviewer’s precise, detailed, and constructive comments and suggestions, after which we have revised the manuscript as described below and in the revised manuscripts
Point 1: The manuscript provides too many details about the development of Deep Learning. It’s better to cut down it to a brief introduction.

Response 1: Thank you for your comment and suggestion. Following your recommendation, we have summarized into two paragraphs in the revised manuscript.
Point 2: In chapter 3.3 Training by the CNN, 928 samples were used to train the model. No augmentation strategies were mentioned. Considering the size of the CNN. The number of the samples may be insufficient.
Response 2: In this paper, the number of sample is limited due to the quality availability in the database. We will suggest augment strategy and increase the number of samples in the future work. We mentioned this in the conclusion and future works paragraph in the revised manuscript.
Point 3: The analysis of the results shown in Figure 11 is sufficient. The straight lines occurred soon after training may be caused by insufficient training samples.
Response 3: Thank you for your comment and suggestion. Following your recommendation, we agree that the result was due to insufficient number of samples, and we will proceed with the experiment by adding more data in addition to the dermatoscope images and diagnostic results provided by ISIC
Point 4: There are no evaluation results on different data sets, and no comparison with other state-of-the-art methods.
Response 4: In this paper, we focused on the evaluation of the suggest algorithm and method. In the future work, we will compare the algorithm with other state-of-the-art methods.
Point 5: Chapter 2.2 Preprocessing does not mention, when performing zero padding, what’s the size of the blank boundary.
Response 5: Thank you for your comment and suggestion. Following your recommendation, the zero padding was set so that the size of the data was 170×170 by adding the same size in each direction as when applying the symmetric padding in Figure.4.
Point 6: It’s not clear why the lesion area in Figure 6(a) and (c) looks different.
Response 6: Thank you for your comment and suggestion. Following your recommendation, as a result of checking the comments, we found a problem during the resizing process and corrected the figure by solving the problem.

Reviewer 2 Report
The authors propose a deep learning-based classification method for malignant melanoma and benign skin tumors from 12 RGB channel skin images.
Page 3, 4, and 5
- The description of machine learning is a little too wordy. Maybe one or two paragraphs would be fine.
Page 6
- The border of the image in Figure 3 does not appear in Figure 4.
- In Figure 4, the size of the yellow box is not defined.
- Isn’t the resizing affected the outcome (classification results)?
Page 7
- In Equation 1, variables in the equation are not defined. Is k prime the same as k?
- Line number 275, w(X) is not defined. Is X lowercase?
- Line number 279, what is the uppercase X? Is it Chi?
Page 8
- Are there any reasons why those hyper-parameters are selected?
Page 9
- In Figure 5, the size of the input/output images is not defined.
- In Figure 5, the size of the images over the U-Net is not defined.
Page 10
- In Figure 6, can’t an expert have an inter/intra observer variation?
- In Figures 6, 7, and 8, it is very difficult for me to compare the annotated area with the predicted area intuitively because of the different locations and sizes of the images.
Page 11
- In Figure 9, what is the difference between (b) and (c)?
- Need to clearly describe why the authors use CNN?
Page 12
- Why is the number of channels in activatioin_12 2? Are they for the benign and malignant cases?
Page 13
- The accuracy is not good enough (around 80 over 100). There are many other metrics to measure the performance of the CNN (i.e. ROCAUC)
Author Response
Response to Reviewer 2 Comments
We are grateful for the reviewer’s precise, detailed, and constructive comments and suggestions, after which we have revised the manuscript as described below and in the revised manuscripts
Point 1: Page 3, 4, and 5. The description of machine learning is a little too wordy. Maybe one or two paragraphs would be fine.

Response 1: Thank you for your comment and suggestion. Following your recommendation, we have summarized two paragraphs.
Point 2: Page 6. The border of the image in Figure 3 does not appear in Figure 4. In Figure 4, the size of the yellow box is not defined. Isn’t the resizing affected the outcome (classification results).
Response 2: Thank you for your comment and suggestion. Following your recommendation,
Figure 3 shows samples of special cases to show samples of our various datasets, and Figure 4 takes and applies general samples to explain the effect of using symmetric padding and zero padding. And the original images are varied and have high resolution. We scaled it to a resolution of 128x128 and the yellow box represents the size of the scaled image 128x128.
Point 3: Page 7. In Equation 1, variables in the equation are not defined. Is k prime the same as k? Line number 275, w(X) is not defined. Is X lowercase? Line number 279, what is the uppercase X? Is it Chi?
Response 3: Thank you for your comment and suggestion. Following your recommendation, we have added definition of variables in equation 1. As where ak(x) denotes the activation in feature channel k at the pixel position. K is the number of classes and pk(x) is the approximated maximum function. Also, w(X) in Line 279 refers to ω(x) in equation 2. We apologize for not clearly marking it, and that part has been corrected.
Point 4: Page 8. Are there any reasons why those hyper-parameters are selected?
Response 4: Thank you for your comment and suggestion. Following your recommendation, we adjusted the hyper-parameters of the model through the validation set, and as a result, the hyper-parameters shown in Table 1 were selected. After applying the selected hyper-parameters, the final effect of the model was confirmed through the test set.
Point 5: Page 9. In Figure 5, the size of the input/output image is not defined. In Figure 5, the size of the images over the U-Net is not defined.
Response 5: Thank you for your comment and suggestion. Following your recommendation, this is described in paragraph 2.5, image output by the U-Net model, the input images were set to 170×170 in size, considering the size of the output images for that of the input images. The dermoscopy images with the size of 170×170, which were pre-processed and output, were also processed by U-Net to be output in the size of 128×128.
Point 6: Page 10. In Figure 6, can’t an expert have an inter/intra observer variation? In Figure 6, 7, and 8, it is very difficult for me to compare the annotated area with the predicted area intuitively because of the different locations and sizes of the images.
Response 6: Thank you for your comment and suggestion. Following your recommendation, as a result of checking the comments, we found a problem during the resizing process and corrected the figure by solving the problem.
Point 7: Page 11. In Figure 9, what is the difference between (b) and (c)? Need to clearly describe why the authors use CNN?
Response 7: Thank you for your comment and suggestion. Following your recommendation, (b) and (c) are the results of different kernel values ​​in order to confirm that the feature map for an image can appear in various ways depending on the value of the kernel applied to the image. In addition, by using CNN, extra-lesion noise was removed to assist the expert in reading.
Point 8: Page 12. Why is the number of channels in activation_12 2? Are they for the benign and malignant cases?
Response 8: Thank you for your comment and suggestion. Following your recommendation, the number of channels of activation 12 is designed to discriminate between benign and malignant, respectively.
Point 9: Page 13. The accuracy is not good enough (around 80 over 100). There are many other metrics to measure the performance of the CNN (i.e. ROCAUC).
Response 9: Thank you for your comment and suggestion. Following your recommendation, the purpose of this study was to develop a primary screening method for early detection of malignant melanoma. To this end, we designed a process that processes both lesion segmentation and discrimination simultaneously, and it was recognized that the segmentation process was satisfactory, but had a rather low accuracy in the discrimination process. To this end, we will increase the accuracy by applying a model with a deeper layer of the neural network.

Round 2
Reviewer 1 Report
The manuscript proposes a deep learning based method for malignant melanoma diagnosis. U-Net is applied to segment the lesion area, and CNN is applied for classification. Training and evaluation were performed on a data set with 1161 samples.
Major comments:
- As the authors write in ‘Chapter 4. Conclusion’, Several researchers have developed models that exhibit satisfactory performance for 340 analyzing malignant melanoma based on deep learning. The authors should cite and introduce those related works in ‘Chapter 1. Introduction’.
- The analysis part of ‘Chapter 3.4. CNN Training Results and Analysis’ is still far from sufficient. The authors should explain the results and limitations in detail.
Minor comments:
- Please double check the format of the figures, e.g.,
Figure 10, left and upper boundary of the figure is missing.
Author Response
We are grateful for the reviewer’s precise, detailed, and constructive comments and suggestions, after which we have revised the manuscript as described below and in the revised manuscripts
Point 1: As the authors write in ‘Chapter 4. Conclusion’, several researchers have developed models that exhibit satisfactory performance for 340 analyzing malignant melanoma based on deep learning. The authors should cite and introduce those related works in ‘Chapter 1. Introduction’.
Response 1: Thank you for your comment and suggestion. Following your recommendation, we have added an introduction to the paper on the diagnosis of malignant melanoma using deep learning in the Chapter 1. Introduction in the revised manuscript.
Point 2: The analysis part of ‘Chapter 3.4. CNN Training Results and Analysis’ is still far from sufficient. The authors should explain the results and limitations in detail.
Response 2: We appreciate your comments and suggestions, and we have additionally written about limitations on training results and directions for improvement in the revised manuscript.
Point 3: Please double check the format of the figures, e.g.,
Response 3: Thank you for your comment and suggestion. Following your recommendation, we checked the format of all figures and modified them to be uniformly the format suggested by the journal.
Point 4: Figure 10, left and upper boundary of the figure is missing.
Response 4: Thank you for your comment and suggestion. Following your recommendation, as can be seen from the text, we used padding in the region segmentation process using U-net. However, in the classification process we use figures of segmented data by U-Net as input data. Therefore, the input data is expressed as 128x128 in the figure.
Reviewer 2 Report
The authors answered my questions and suggestions, but they didn't improved the evaluation part. Although it is a bit critical part in another discipline, I feel that the idea of this paper is good enough to be accepted in the journal. Furthermore, they said that they will improve it using another model. Hope to verify their approach soon.
Author Response
We are grateful for the reviewer’s precise, detailed, and constructive comments and suggestions, after which we have revised the manuscript as described below and in the revised manuscripts
Point 1: The authors answered my questions and suggestions, but they didn't improved the evaluation part. Although it is a bit critical part in another discipline, I feel that the idea of this paper is good enough to be accepted in the journal. Furthermore, they said that they will improve it using another model. Hop e to verify their approach soon.

Response 1: First of all, we are very grateful for these comments and suggestions. We will submit a more advanced paper by adding the suggestions in the future. Thanks again for your detailed comments and suggestions.
